# The Effect of the COVID-19 Pandemic on Unrelated Allogeneic Hematopoietic Donor Collections and Safety

Gaganvir Parmar [1,2], David S. Allan [1,3], Gail Morris [1], Nicholas Dibdin [1], Kathy Ganz [1], Karen Mostert [1], Kristjan Paulson [4,5], Tanya Petraszko [1,6], Nora Stevens [1] and Matthew D. Seftel [1,6,*]

1 Stem Cells, Canadian Blood Services, Ottawa, ON K2E 8A6, Canada
2 Faculty of Medicine, University of Toronto, Toronto, ON M5S 1A8, Canada
3 Department of Medicine and Biochemistry, Microbiology & Immunology, Faculty of Medicine, University of Ottawa, Ottawa, ON K1H 8M5, Canada
4 Cell Therapy and Transplant Canada, Winnipeg, MB R3P 2R8, Canada
5 Department of Internal Medicine, Max Rady College of Medicine, University of Manitoba, Winnipeg, MB R3E 0V9, Canada
6 Department of Medicine, Faculty of Medicine, University of British Columbia, Vancouver, BC V1Y 1T3, Canada
* Correspondence: matthew.seftel@blood.ca; Tel.: +604-707-3414

**Abstract:** Background and Objectives: The COVID-19 pandemic profoundly influenced unrelated donor (UD) allogeneic peripheral blood stem cell (PBSC) collections. Changes included efforts to minimize COVID-19 exposure to donors and cryopreservation of products. The extent to which the efficacy and safety of PBSC donations were affected by the pandemic is unknown. Methods: Prospective cohort analysis of PBSC collections comparing pre-pandemic (01 April 2019–14 March 2020) and pandemic (15 March 2020–31 March 2022) eras. Results: Of a total of 291 PBSC collections, cryopreservation was undertaken in 71.4% of pandemic donations compared to 1.1% pre-pandemic. The mean requested CD34$^+$ cell dose/kg increased from $4.9 \pm 0.2 \times 10^6$ pre-pandemic to $5.4 \pm 0.1 \times 10^6$ during the pandemic. Despite this increased demand, the proportion of collections that met or exceeded the requested cell dose did not change, and the mean CD34$^+$ cell doses collected ($8.9 \pm 0.5 \times 10^6$ pre-pandemic vs. $9.7 \pm 0.4 \times 10^6$ during the pandemic) remained above requested targets. Central-line placements were more frequent, and severe adverse events in donors increased during the pandemic. Conclusion: Cryopreservation of UD PBSC products increased during the pandemic. In association with this, requested cell doses for PBSC collections increased. Collection targets were met or exceeded at the same frequency, signaling high donor and collection center commitment. This was at the expense of increased donor or product-related severe adverse events. We highlight the need for heightened vigilance about donor safety as demands on donors have increased since the pandemic.

**Keywords:** allogeneic hematopoietic cell transplantation; COVID-19; cryopreservation; peripheral blood hematopoietic stem cells

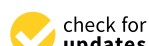



## 1. Introduction

Haematopoietic cell transplantation (HCT) offers the potential to cure patients with hematologic malignancies, bone marrow failure syndromes and inherited metabolic diseases [1]. Canadian Blood Services (CBS) coordinates unrelated donor (UD) peripheral blood stem cell collections (PBSC) in partnership with hospital-based apheresis centres for patients in Canada and internationally [2].

PBSCs remain the most common source of cells collected from UDs for HCT owing to increased convenience, rapid rates of engraftment and the avoidance of surgical procedures for donors [3]. Prior to PBSC collection, transplant centers, collection facilities, and CBS collectively agreed on CD34$^+$ cell dose collection targets that would be safe for the donor

and clinically effective for the recipient. Although the optimal cell dose for allogeneic HCT remains a topic of active research [4], collection centers often set minimum targets of $2 \times 10^6$ CD34$^+$ cells/kg recipient weight [5], while $\geq 5 \times 10^6$ CD34$^+$ cells/kg is often considered optimal [6]. Infusing very high doses of total cells has been associated with adverse patient outcomes [7,8], and efforts to collect such doses may increase adverse events or strain collection centre capacity by increasing the likelihood of requiring additional days of apheresis [9].

The COVID-19 pandemic introduced additional challenges to PBSC collections from UDs. Complex and uncertain transportation logistics threatened the safety of fresh delivery of products for recipients. Transplant centers adapted by modifying their selection approach for donors [10], and numerous transplantation professional bodies suggested cryopreservation of collected products to grant flexibility in coordinating infusions at a later date [11,12]. Previous research demonstrated a decrease in CD34$^+$ cell quantities and viability from cryopreserved products [13,14], but the impact of pandemic donor collection and product cryopreservation practices on requested cell doses, the ability of collection centres to adapt to these changes and donor safety is unknown. Furthermore, the impact of efforts to minimize COVID-19 exposure risk to donors during PBSC collection, such as the reduced need for donors to travel long distances to collection sites and limiting collections to a single day, is also unknown. In this study, we investigated the extent to which PBSC donations were impacted by the COVID-19 pandemic.

## 2. Methods

Prospectively collected data on consecutive PBSC collections coordinated by CBS between 1 April 2019, and 1 April 2022, were extracted from the CBS electronic database. Results were analyzed by comparing pre-pandemic (1 April 2019–14 March 2020) and pandemic (15 March 2020–31 March 2022) PBSC collection cohorts. Serious (Product) Events and Adverse Reactions (S(P)EAR) related to donors or PBSC products were extracted from CBS reports that had been submitted to the World Donor Marrow Association (WDMA) Serious Events and Adverse Reactions system. All data were de-identified and aggregated prior to analysis.

CBS utilizes a total of seven collection centres, all based at established, high-volume (>100 transplants per year) HCT centres. Three of these are based in the largest province of Ontario, with one each in the provinces of Nova Scotia, Manitoba, Alberta, and British Columbia. Since PBSC collection centres are not present in every province and territory of Canada, the significance of out-of-province donation travel was characterized by dividing Canada into three regions: a "central region" containing the provinces of Ontario and Quebec; a "western region" containing the provinces of British Columbia, Alberta, Saskatchewan, Manitoba, and the northern territories; an "eastern region" containing the Atlantic provinces of New Brunswick, Nova Scotia, Prince Edward Island, and Newfoundland and Labrador. Intra-regional long-distance travel was defined as out-of-province/territory travel but still within one of these three regions. Inter-regional long-distance travel entailed travel from one region to another of these three regions.

Statistical analyses were performed using Prism 9 (GraphPad) or Microsoft Excel software (v16.0.6366.2062). Statistical significance for comparisons was performed using either Student's *t*-test for numerical data or Fisher's exact test for categorical data. Data are shown as means $\pm$ SEM when applicable. Statistical significance was accepted at $p < 0.05$, with $p$ values listed or represented with the following denominations: *** = $p < 0.001$, ** = $p < 0.01$, * = $p < 0.05$. Graphs and visuals were generated using GraphPad Prism 9 software.

## 3. Results

A total of 292 PBSC collections were conducted during the study period (1 April 2019 to 31 March 2022). There were 95 collections prior to 15 March 2020 (8.26 collections per month) and 197 (8.04 per month) thereafter (incidence rate ratio 1.01, $p = 0.9$). One pandemic-era donation collected from an adult donor for an infant patient who weighed

only 9 kg was considered an outlier and was excluded from the subsequent analysis. Donor demographic and characteristics were similar before and during the pandemic (Table 1). Regarding donor travel beyond the donors' home province/territory, there was a reduced proportion of this long-distance travel during the pandemic (9.2%) compared to pre-pandemic (16.8%, $p = 0.07$). There were no appreciable differences in donor gender or ethnicity comparing the pre-pandemic and pandemic eras.

**Table 1.** Characteristics of Allogeneic PBSC Transplant Unrelated Donors.

| | All Donors (N = 291) | Pre-Pandemic (N = 95) | Pandemic (N = 196) | *p*-Value |
|---|---|---|---|---|
| **Donor Age (*N*, (%))** | | | | |
| 18–25 | 117 (40.2) | 38 (40) | 79 (40.3) | 0.99 |
| 26–35 | 133 (45.7) | 41 (43.2) | 92 (46.9) | 0.62 |
| 36–45 | 33 (11.3) | 11 (11.6) | 22 (11.2) | 0.26 |
| 46–55 | 6 (2.1) | 4 (4.2) | 2 (1) | 0.09 |
| 56–65 | 2 (0.7) | 1 (1.1) | 1 (0.5) | 0.55 |
| 66+ | 0 (0) | 0 (0) | 0 (0) | - |
| Median | 27 | 28 | 27 | - |
| Mean ± SEM | 28.4 ± 0.4 | 29 ± 0.9 | 28 ± 0.5 | 0.29 |
| **Donor Sex (*N*, (%))** | | | | |
| Male | 213 (73.2) | 73 (76.8) | 140 (71.4) | 0.40 |
| Female | 78 (26.8) | 22 (23.2) | 56 (28.6) | 0.40 |
| **Donor Ethnicity (*N*, (%))** | | | | |
| Caucasian | 230 (79) | 72 (75.8) | 158 (80.6) | 0.36 |
| Other | 61 (21) | 23 (24.2) | 38 (19.4) | 0.36 |
| **Out-of-province/territory travel (*N*, (%))** | | | | |
| Inter-region | 15 (5.2) | 6 (6.3) | 9 (4.6) | 0.58 |
| Intra-region | 19 (6.5) | 10 (10.5) | 9 (4.6) | 0.07 |
| Total | 34 (11.7) | 16 (16.8) | 18 (9.2) | 0.07 |

SEM: standard error of the mean.

PBSC collection characteristics are summarized in Table 2 and Figure 1. We observed an increase in the CD34$^+$ cell doses/kg recipient weight that were requested during the pandemic ($p = 0.01$). However, there was no difference in the actual cell dose collected ($p = 0.29$). Multi-day collections were numerically more frequent in the pandemic era (2.1% pre-pandemic vs. 5.1% pandemic, $p = 0.35$).

**Table 2.** Pre-Pandemic and Pandemic Allogeneic PBSC Collection Characteristics.

| | Pre-Pandemic (N = 95) | Pandemic (N = 196) | *p*-Value |
|---|---|---|---|
| Requested CD34+ dose ($\times 10^6$ cells) | 368.5 ± 10.6 | 393.3 ± 9.4 | 0.11 |
| Requested dose CD34+ dose ($\times 10^6$ cells/kg recipient weight) | 4.9 ± 0.2 | 5.4 ± 0.1 | 0.01 |
| Collected CD34+ cell dose ($\times 10^6$ cells) | 650.6 ± 34 | 672.2 ± 22.3 | 0.61 |
| Collected CD34+ cell dose ($\times 10^6$ cells/kg recipient weight) | 8.9 ± 0.5 | 9.7 ± 0.4 | 0.29 |
| Proportion of cell dose requests filled (%) | 88.4 | 87.2 | 0.79 |
| Cryopreserved products (%) | 1 (1.1) | 140 (71.4) | <0.0001 |
| Two-day collections (%) | 2 (2.1) | 10 (5.1) | 0.35 |
| Product not infused (%) | 0 (0) | 6 (3.1) | 0.18 |
| CVAD placement (%) | 1 (1.1) | 12 (6.1) | 0.07 |
| Donor- and Product-related severe adverse event (%) | 0 (0) | 5 (2.6) | 0.17 |

PBSC: peripheral blood stem cell collections; SEM: standard error of the mean; CVAD: central venous access device.

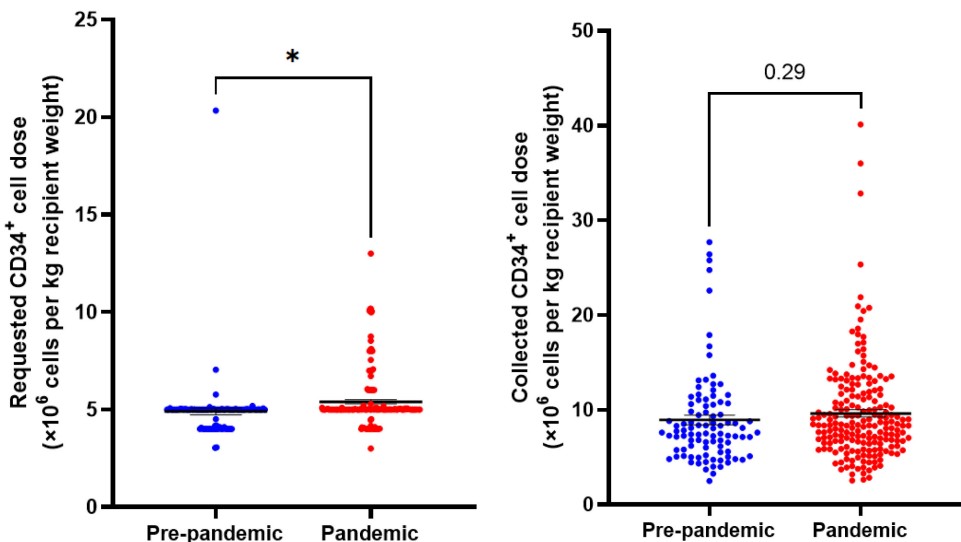

**Figure 1.** Requested and collected CD34$^+$ cell doses per kg recipient weight in pre-pandemic and pandemic collections. $N = 291$. Means indicated $\pm$ SEM. * = $p < 0.05$.

The frequency of cryopreserving PBSC collections increased during the pandemic from 1.1% to 71.4% ($p < 0.0001$). Moreover, cryopreservation frequency during the pandemic increased from year 1 to year 2 (59/98 (0.2%) vs. 81/98 (82.7%), $p = 0.0008$). There were more non-infused donations during the pandemic ($N = 6$ (3.1%)) compared to none pre-pandemic. All non-infused donations during the pandemic were in cryopreserved PBSC products. However, these non-infusion events were not in those donations that failed to meet the requested cell dose. The most frequent reason for non-infusion of the collected product was patient death that had occurred after the PBSC collection but prior to the planned PBSC infusion (3/6, (50%)).

Regarding adverse events, there were no donor or product severe adverse events (SAEs) reported to the WMDA S(P)EAR system during the pre-pandemic period. In contrast, there were four donor-related SAEs and one product-related SAE during the pandemic (Table 3). All donor-related SAEs occurred in cryopreserved PBSC collections, while the product-related SAE was not cryopreserved. There was an increase in the number of donors who underwent insertion of a central venous access device during the pandemic (1/95 (1%) vs. 12/196 (6.1%), $p = 0.07$). None of the observed SAEs were directly related to the insertion or presence of a CVAD.

**Table 3.** Donor or Product Severe Adverse Events.

| SAE Type | Age, Sex | Weight (kg) | GCSF Dose (mcg/kg/day) | Event Description |
|---|---|---|---|---|
| Donor | 40 F | 74 | 8 × 4 days | Donor reported sternal pain radiating to back and headache following GCSF administration. Morphine taken for pain which caused nausea and vomiting. Donor admitted to hospital the night before apheresis and was able to proceed with collection the following day. Lab tests showed elevated liver enzymes. |
| Donor | 32 M | 86 | 10.5 × 4 days | Citrate reaction during donation. Peripheral numbness was noted, which resolved with calcium. Patient was discharged with no symptoms, but the following day symptoms recurred and persisted for 6 days. Paresthesia is slowly resolving. |
| Donor | 21 M | 75 | 10 × 4 days | Donor developed acute pancreatitis 105 days following donation. Gastrointestinal symptoms started 3 weeks post-donation. |
| Donor | 26 F | 63 | Not reported | Donor tested positive for COVID-19 two days after collection. Donor reported fever, sinus pain, headache, bone and muscle pains, light-headedness. Fever and chills started 1-day post apheresis. |
| Product | - | - | - | Product labelling error leading to PBSC allocation to an incorrect recipient. The recipient recovered fully. |

SAE: Severe Adverse Event; GCSF: Granulocyte colony-stimulating factor; F: Female; M: Male; PBSC: Peripheral Blood Stem Cells.

## 4. Discussion

The continued support, trust, and safety of volunteer UDs remain crucial to maintaining access to PBSCs for allogeneic HCT, especially in the context of a pandemic. While the need for HCT continued, access to the ability to collect cells from UDs remained robust over the first two years of the pandemic. Although requested cell doses increased in the setting of planned cryopreservation, collection centres were able to meet these increased targets with the same frequency as during the pre-pandemic period. Maintaining the capacity of collection centres and the goodwill of UDs will be key factors in overcoming future threats to donor collections.

Despite a brief initial decrease in UD demand in Canada during the first six months of the pandemic [2], overall usage of UDs from the CBS Stem Cell Registry has remained consistent, as evident by the stable number of UDs undergoing PBSC collections in the year before compared to the first two years of the pandemic.

During the pandemic, we attempted to minimize donor COVID-19 exposures. One such way was to reduce travel requirements for donors [10]. In accordance with this effort, we noted a drop in the frequency of long-distance travel for the purpose of donations. This highlights the ongoing commitment of CBS and collections centres to provide PBSCs closer to donors' places of residence. This also emphasizes the need for long-term expansion of accessible, local collection facilities, which in turn is likely to improve convenience and lower infection-related exposures for donors.

Cell dose requests, adjusted for recipient weight, increased by approximately 10% during the pandemic. This was in association with the increased use of cryopreservation. A recent observational study analyzing the first six months of the pandemic reported an 8% reduction in the median infused CD34+ cell doses with cryopreserved products [15]. This observation may account for the observed rise in requested cell dose in our analysis as transplant centres tried to offset anticipated cell losses during cryopreservation. Despite an increase in requested cell doses during the pandemic, PBSC collections still exceeded the requested cell dose in most cases. This suggests that collection centers can respond to the increase in cell dose requests that are related to planned cryopreservation.

We observed higher rates of unplanned or adverse events in donors during the pandemic, with a higher number of two-day collections and central venous access device placements. Although the absolute number of SAEs was low, the frequency of these events was higher during the pandemic. It is plausible that the higher cell dose demands placed on collection centres and donors during the pandemic led to these unplanned or adverse

events. Another explanation for the higher number of unplanned or adverse events in donors or products is that HCT collection centres may have been negatively affected by new or additional tasks that were required during the COVID-19 pandemic. Increased demands, especially during the complex process of coordinating an unrelated donor HCT, may lead to increased error rates, as described in other medical practice areas [16]. However, the small number of observed events makes this theory tentative, and we recommend that further study about this preliminary observation is warranted.

Whether planned cryopreservation will continue in the future remains uncertain. Despite initial concerns that SARS-CoV2 could be transferred from a COVID-19-positive donor via their hematopoietic cell product, reassuring data emerged during the pandemic that this was not the case. As these concerns about COVID-19 blood-borne transmission subsided [17,18], we observed that cryopreservation frequency did not drop accordingly. It is likely that the popularity of cryopreservation was maintained as HCT teams have become more familiar and confident about the safety and feasibility of this precautionary practice. However, if cryopreservation were to continue as a standard element of PBSC donation, we are concerned that this may lead to a consistent trend toward higher donor-associated adverse events. We and other registries are viewing cell dose requests with greater scrutiny in the hope of preventing potential inconvenience to, discomfort in, and adverse effects on donors. We are also concerned about the potential for the collection and storage of products that are never infused, as this represents an unnecessary donation by a volunteer donor. Moreover, an earlier COVID-19 study has previously linked the mortality of HCT patients to delayed infusion of cryopreserved products [19], adding a further pause to the practice of widespread cryopreservation.

Although optimal PBSC doses are disease and treatment dependent, mean collected doses in this study ($8.9 \pm 0.5 \times 10^6$ CD34$^+$ cells/kg pre-pandemic vs. $9.7 \pm 0.4 \times 10^6$ CD34$^+$ cells/kg pandemic) are aligned with doses suggested in recent HCT research [4,6,8]. Lower cell doses collected in previous studies, in conjunction with cell losses due to cryopreservation, may have contributed to the inferior outcomes associated with cryopreservation in earlier pre-pandemic research [13,14,20]. However, a recent CIBMTR analysis of PBSC donations during the first six months of the pandemic found no differences in overall survival and engraftment following the infusion of cryopreserved versus fresh products [15]. Further studies are needed to investigate whether the widespread adoption of cryopreservation in COVID-19 is associated with any short- or long-term adverse outcomes in recipients and whether CD34$^+$ cell doses are associated with such outcomes.

This study has several limitations. The small number of PBSC collections and observed adverse events resulted in lower statistical power, and we recommended that our study be replicated using a larger donor registry dataset. However, its design as a national-level, prospective cohort study with detailed collection-related data add to its validity. We reported severe donor- or product-related events over the study period but did not report on minor adverse events; although this would have been an ideal addition to the paper, minor adverse events are inconsistently reported to the donor registry, which would affect the reliability of these additional results. We thus elected to report only on severe adverse events. Our study did not include bone marrow harvests from unrelated donors; this procedure is undertaken much less frequently, and it results in a distinct experience for donors and collection centres. The characteristics and outcomes after bone marrow harvest will be the subject of a separate analysis by our registry.

In conclusion, the ability to perform successful PBSC collections from UDs remained robust in Canada during the COVID-19 pandemic. Donor willingness and motivation, coupled with robust collection capacity, was able to meet increased cell targets requested by transplant centres to support the widespread cryopreservation of products. However, the increase in unplanned or adverse events in PBSC donors serves as a reminder to exercise increased attention to donor safety during the pandemic, especially as planned cryopreservation of PBSC products continues.

**Author Contributions:** Conceptualization, M.D.S. and D.S.A.; methodology, G.P., M.D.S. and D.S.A.; validation, G.P., K.M., N.D., K.G. and T.P.; formal analysis, G.P.; investigation, G.P., G.M., M.D.S. and D.S.A.; resources, K.G.; data curation, G.M., N.S. and K.M.; writing—original draft preparation, G.P., M.D.S. and D.S.A.; writing—review and editing, all authors including K.P.; visualization, G.P., M.D.S. and D.S.A.; supervision, M.D.S. and D.S.A.; project administration, K.G., N.D., M.D.S. and D.S.A.; funding acquisition, K.G. All authors have read and agreed to the published version of the manuscript.

**Funding:** This study was supported by operational funds at Canadian Blood Services.

**Institutional Review Board Statement:** Ethical review and approval were waived for this study due to the use of anonymized data for the purposes of operational improvement.

**Informed Consent Statement:** Not applicable.

**Data Availability Statement:** The data presented in this study are available on request from the corresponding author.

**Acknowledgments:** We thank all donors of the CBS registry for their commitment and resilience, especially during the COVID-19 pandemic. We also extend our gratitude to the staff at the collection centres that facilitated the collection of cells from these unrelated donors.

**Conflicts of Interest:** All authors are employed by or have received remuneration from Canadian Blood Services.

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
