# Peer review of "The Effect of the COVID-19 Pandemic on Unrelated Allogeneic Hematopoietic Donor Collections and Safety"

_curroncol, doi:10.3390/curroncol30030270_

Round 1

Reviewer 1 Report

This is a study looking into PBSC UD collection pre-pandemic and during pandemic. Study clearly elucidates the objective, design plan, results and conclusion. One thing which can be considered - little elaboration on the observed SPAE in the study so that centers can prepare to minimize or avoid the SPAE. 

Author Response

We are very grateful for this reviewer's insightful critique and suggestion.

In direct response to this reviewer we have elaborated on the observed SPAEs with a new table (table 3), and more detailed description about the AEs in the "results" section of the paper.  Kindly see the revised paper for these additions.

Reviewer 2 Report

The authors submit an interesting analysis about the impact of the COVID19 pandemic on unrelated donor stem cell collection and safety.

Results show that a higher amount of stem cells was often requested by the recipient center and that probably for this reason, a central venous line was more often placed for stem cell collection. The rate of cryopreservation was very high and 6 products were finally nit infused.

The results and the discussion are very interesting; however, some issues remain:

-          Donors were often collected not so far away from their home. Was this also related to collection of stem cells in less experienced donor centers and the reason for the more frequent use of central venous lines?

-          Multi-day collection increased from 2.1 to 5.1 %. This is more than a doubling. Even if the number of events is rather low, is this really not significant? Has the correct statistical approach been used? The same must be checked for the difference in products not infused ( 0 versus 6 events) and placement of CVAD

-          What were exactly the SAE in the donors? Was it CVAD related?

-          Was the frequency of cryopreservation the same during the first compared to the second year of the pandemic?

He authors must discuss more in detail if there was a decrease of the frequency of cryopreservation during the second year. During the pandemic, results from the transplant community have shown that collection of stem cells was also possible in asymptomatic COVID19 stem cell donors and that there was no impact on patients. Indication for cryopreservation decreased during the pandemic and this is also important to avoid futile stem cell collection and to protect stem cell donors.

Author Response

We are very grateful for this reviewer's insightful critique and suggestions. Kindly see the revised paper that incorporates these constructive suggestions.

In direct response to this reviewer:

  1. Donors were often collected not so far away from their home. Was this also related to collection of stem cells in less experienced donor centers and the reason for the more frequent use of central venous lines?

Response: We agree that this matter needs further elaboration.  In the introduction section of the paper, we now describe the number and location of all the collection centres. There are only 7 HCT centres that perform unrelated donor collections, all of which are long established, high volume HCT centres.  Although we agree with this reviewer that centre experience is an important factor with respect to both donor and recipient outcomes, our study was not capable of examining this variable. We have now emphasized in the discussion section the importance of confirming our initial observations in a larger registry dataset. 

2. Multi-day collection increased from 2.1 to 5.1 %. This is more than a doubling. Even if the number of events is rather low, is this really not significant? Has the correct statistical approach been used? The same must be checked for the difference in products not infused (0 versus 6 events) and placement of CVAD.

Response: We agree that these observations are important, yet limited by low event rates and associated statistical power.  To analyze categorical data, contingency tables were created and analyzed using Fisher’s Exact Test. Statistical significance of these low frequency events is likely limited by the scale of the study, which is a limitation. Nonetheless, a doubling in frequency of safety related events, as the reviewer points out, should not be ignored. We have highlighted this  in the results section and amended discussion section. We have also emphasized in the discussion section the important of confirming our initial observations in a larger registry dataset. 

3.  What were exactly the SAE in the donors? Was it CVAD related?

Response: We agree that this is worthy of elaboration. We have added a new table ( table 3) that elaborates on all the observed SAEs.   We also add in the results section that none of the SAEs appeared to be directly related to CVADs.  In the discussion section we now also cite evidence that medical errors in general increased during the COVID-19 pandemic, and that HCT is likely not immune to this risk in the setting of increased, complex work demands.

4. Was the frequency of cryopreservation the same during the first compared to the second year of the pandemic?

Response: We agree that this is worthy of elaboration. We re-examined the dataset, and have added into the results section that cryopreservation in year 1 of the pandemic was  60.2% and increased in year 2 of the pandemic to 82.7%. P-value = 0.0008, Fisher’s Exact Test.  In the discussion section we now theorize as to why this temporal trend has occurred and we also have added relevant references to support this discussion. 

Reviewer 3 Report

The authors have analyzed in their communication the effect of the COVD-19 pandemic on unrelated allogeneic hematopoietic donor collections and safety.

The study is very well done. I want to congrats the authors for the clear method of presenting the results.

The abstract is well written and is setting an idea of the study to a potential reader.

The introduction part is well written and documented.

The methods part is presenting the main objects use to do the study. The periods of the study are well defined. Are there any data which were excluded from the study? Please specify this.

The results are very well presented and in a concise form.

The discussions are emphasizing the results of the study. The conclusion is answering the hypothesis of the study which was set by the title. The limitations of the study are well presented.

In my opinion, the authors should add a few more references to their communication (to have at least 20).

Author Response

We are very grateful for this reviewer's insightful critique and suggestions. Kindly see the revised paper that incorporates these constructive suggestions.

In specific response to this reviewer:

  1. Are there any data which were excluded from the study? Please specify this.

We agree that this point is worthy of elaboration. Donor collection events were from consecutively collected donors over the study period.    However, in this paper we only report on "severe" donor or product related events over the study period. We did not report on "minor" adverse events that occurred; although this would have been an ideal addition to the paper, minor AEs are inconsistently reported to the donor registry, which would affect the reliability of these results.   We thus elected to report only on severe adverse events.   We have added this limitation to the discussion section of the paper. 

2. In my opinion, the authors should add a few more references to their communication (to have at least 20).

Based on the collective suggestions of all reviewers, we have added three additional relevant references to the paper.

Kindly refer to the revised manuscript to see all of our revisions.